# Trade-Off Analyses of Multiple Ecosystem Services and Their Drivers in the Shandong Yellow River Basin

**DOI:** 10.3390/ijerph192315681

**Published:** 2022-11-25

**Authors:** Xufang Zhang, Yu Yang, Minghua Zhao, Rongqing Han, Shijie Yang, Xiaojie Wang, Xiantao Tang, Weijuan Qu

**Affiliations:** 1College of Geography and Environment, Shandong Normal University, Jinan 250358, China; 2College of Earth and Environmental Sciences, Lanzhou University, Lanzhou 730000, China

**Keywords:** ecosystem services, driving factors, trade-offs, ALM, GWR, Shandong Yellow River Basin

## Abstract

With the intensification of conflicts between different ecosystem services, how to achieve a win-win situation between socio-economic development and ecological protection is an important issue that needs to be addressed nowadays. In particular, how to better quantify and assess the intensity of ecosystem service trade-offs and their relative benefits, and to identify the influencing factors are issues that need to be studied in depth. Based on the INVEST model, this paper analyzed the evolution of spatial and temporal patterns of ecosystem services such as Carbon Storage (CS), Food Production (FP), Habitat Quality (HQ), and Water Yield (WY) in the Shandong Yellow River Basin (SYRB) in 2000, 2010 and 2020. Next, we quantitatively measured the trade-off intensity and revealed the key influencing factors of the trade-off intensity evolution using automatic linear models, root mean square deviation, and geographically weighted regressions. Subsequently, we further analyzed the impact of the correlation between environmental and socio-economic factors on the trade-off intensity of ecosystem services. The results indicated that the temporal and spatial changes of the four main ecosystem services in SYRB area were inconsistent. WY showed a fluctuating trend, with a large interannual gap. CS and FP are on the rise, while HQ is on the decline. Spatially, WY and HQ showed a decreasing distribution from the center to the periphery, while FP and CS showed a decreasing distribution from the southwest to the northeast. The location characteristics of SYRB’s four ecosystem services and their trade-offs were obvious. FP had absolute location advantage in ecosystem service trade-offs. Most of the four ecosystem services showed significant trade-offs, and the trade-off intensity had significant spatial heterogeneity, but the trade-off between FP and CS was relatively weak. At the same time, there were also differences in the trends of trade-off intensities. Counties with low trade-off intensity were mostly located in mountainous areas; these areas are less disturbed by human activities, and most of them are areas without prominent services. Counties with high trade-off intensities were mostly concentrated in areas with relatively developed agriculture; these areas are more disturbed by human activities and are mostly prominent in FP. The trade-off intensity of ecosystem services in SYRB was affected by several factors together, and there were difference characteristics in the degree and direction of influence of each factor. Moreover, these influencing factors have gradually changed over 20 years. In terms of the spatial distribution at the county scale, the influence areas of the dominant drivers of different trade-off types varied greatly, among which the areas with NDVI, CON, and PRE as the dominant factors were the largest. In the future, in effectively balancing regional economic development and ecological environmental protection, quantifiable correspondence strategies should be developed from the administrative perspective of counties and regions based on comprehensive consideration of the locational advantages of each ecosystem service and changes in trade-offs.

## 1. Introduction

Ecosystem services (ESs) are the direct and indirect benefits that humans derive from natural ecosystems [1], which mainly include provisioning services, supporting services, regulating services and cultural services, and are crucial in current landscape optimization and ecosystem management [2,3]. Due to the rich types of ecosystem services, the heterogeneity of their spatial distribution and the human demand preference for ecosystem services, a complex and close non-linear relationship has been formed between ecosystem services, including a synergistic relationship that has been strengthened or weakened at the same time [4] and a trade-off relationship that the enhancement of one service will inhibit the supply of other ecosystem services [5,6], thus making the aims of service diversification and benefit maximization difficult to fulfill. Specially, trade-offs do not only occur in the form of unidirectional changes with uneven rates [7], as well as the ebb and flow of changes, but can also have synergistic effects of mutual benefit [8]. Therefore, exploring the strength of trade-offs in ecosystem services and the mechanisms that govern them is a critical problem for ecosystem services and regional sustainable development [9,10]. Policymakers have also long been seeking effective strategies to modify trade-offs between ESs to maximize the required ESs [11], and how to identify the main drivers affecting ESs’ relationships is crucial to making relevant decisions [12]. Since the 1970s, approximately 60% of global ecosystem services have been in decline with climate change and environmental disturbances from human activities, and some ecosystem service management measures are urgently needed [13]. So, conducting research on ecosystem service, especially with a focus on ecosystem service trade-offs, is important for maintaining ecological balance and improving the overall benefits of ESs under the background of the continuous growth of human activity intensity and material demand [14].

Scholars have made great progress in the study of ecosystem services in many aspects. The current research mainly analyzes the ecosystem service valuation for specific regions and the changes among various ecosystem services through a variety of methods, and reveals the impact mechanism of ecosystem service trade-off relationship through comparative analysis with the spatial and temporal characteristics [14,15], revealing the important role of each influencing factor on the tradeoff relationship, so as to clarify the adjustment methods and strategies of the trade-off relationship. From the perspective of research methods, some of them include the systematic mapping method [8], Spearman rank correlation analysis method [15], Bayesian belief network semi quantitative model [16], InVEST model (integrated valuation of ecosystem services and trade-offs) [17], CASA model (Carnegie–Ames–Stanford approach) [18], and ARIES model (artificial intelligence for ecosystem services) [19]. In the research area, there are not only studies carried out on the global scale, but also studies on urban areas [20], agricultural production areas [21], economically developed areas [22] or specific water source areas [23]. In addition, the scale of research on ecosystem service trade-offs is diverse; the grid scale is determined by combining data resolution or experience, a too-fine township unit scale is adopted, or a too-coarse prefecture-level or even provincial-level scale is adopted, so it is impossible to formulate targeted regional ecosystem management policies. However, there are few quantitative analysis on the impact of natural and human activities on the dynamic change of ecosystem service trade-off relationships [24], and the correlation analysis between environment and socio-economic factors is not deep enough [25,26]. Some studies have adopted spatial analysis units that are not conducive to regional designation of future responses, and others have ignored the variation of regression parameters with geographic location when exploring the influencing factors, which leads to certain limitations in the understanding of ecosystem service trade-off relationships and its driving mechanism. Accordingly, it is also impossible to formulate targeted regional ecosystem management policies. However, geographically weighted regression models (GWR) can identify the local characteristics of variables [27] and thus analyze the spatial relationships between ecological processes. A growing number of studies have also demonstrated the ability of this model to address these issues [28]. In addition, a correct scientific perception, and understanding and weighing of the interrelationships among ecosystem services at a specific scale is beneficial for local governments to make targeted regional ecosystem management decisions and achieve the sustainable development of regional ecosystem services. In China, the general policy implementation unit is mainly at the county level, and the conclusions of different research scales should be summarized to the county level finally.

Studies between ecosystem services conducted in the Yellow River Basin of China are more typical [29,30,31,32]. The Yellow River basin has a fragile ecosystem and high intensity of human activities, especially in the middle and lower reaches, where, on the one hand, the population is dense and socio-economically developed, and on the other hand, large areas of the ecosystem are continuously degraded and form a fragmented landscape [33]. In addition, the key constraints in the development of many countries and cities are related to resources, environment and ecology [34]. Like many other regions, the Yellow River Basin (YRB), has also faced a variety of ecological and environmental issues over the course of societal civilization [33,35]. In 2019, China elevated ecological protection along the Yellow River Basin (YRB) to a national strategy [36]. In 2020, Shandong provincial government took the lead in carrying out YRB ecological protection work. In particular, the various Yellow River diversion projects implemented are an important barrier for regional ecological security, and also the key to downstream ecological protection and flood control and disaster reduction, which are particularly conducive to regional ecological protection and high-quality development. Shandong is one of the major grain-producing provinces in China, and the Shandong Yellow River Basin (SYRB) is a major grain-producing area in Shandong Province, with rich arable land resources and water and heat conditions, and relatively strong grain production capacity. Located in the lower reaches of the Yellow River and connected to the Bohai Sea in the north, the SYRB is rich in water resources, densely populated, and has developed industries and agriculture, which require a large amount of water resources. Although many studies have been conducted on the ESs of the middle and lower reaches of the Yellow River [37,38], most of them ignore the trade-offs between ESs at the administrative scale, which is the basic unit of ecological management decision-making. In addition, studies on the influence of drivers on the spatial heterogeneity of trade-offs among ESs are not sufficiently developed [39]. If spatial heterogeneity is not considered, this may lead to a less accurate analysis of the influence of relevant factors on the relationships between various types of ecosystem services, which in turn may make the identified decisions regarding ecosystem service management flawed.

In this context, this study focused on the Shandong Yellow River Basin in China and selects four ecosystem service types, including Food Production services (FP), Water Yield services (WY), Habitat Quality (HQ), and Carbon Storage (CS), to explore the spatial and temporal distribution and evolution of ESs in SYRB in 2000, 2010 and 2020; at the same time, we further analyzed the intensity of change in trade-offs due to different directions and rates of change among ecosystem services and summarize the relative benefits of trade-offs among ecosystem services at the county level, which is the basic scale for ecological management decisions in China. Finally, the main driving factors affecting ecosystem service trade-offs in different regions were explored, and the impact mechanism of natural, socio-economic and other factors on the trade-off intensity are revealed.

## 2. Materials and Methods

### 2.1. Data Sources

The data used in this study mainly include land-use data, digital elevation data (DEM), administrative boundaries, precipitation, evapotranspiration, and socio-economic data, and soil-type data, spanning the period 2000–2020. Among them: (1) land use data for the study years are obtained from the Resource and Environmental Science Data Centre of the Chinese Academy of Sciences at a resolution of 30 m for six primary land categories and 25 secondary land categories (https://www.resdc.cn (accessed on 5 October 2021)); (2) DEM data are obtained from the Geospatial Data Cloud at a resolution of 1000 m (https://www.gscloud.cn/ (accessed on 8 May 2022)); (3) precipitation and evapotranspiration data are obtained from the Resource and Environment Science Data Centre of the Chinese Academy of Sciences (http://www.geodata.cn/ (accessed on 5 October 2021)); (4) NDVI data are obtained from the Resource and Environment Science Data Centre of the Chinese Academy of Sciences, with a resolution of about 500 m (http://www.geodata.cn/ (accessed on 5 October 2021)); (5) grain production data are obtained from the Shandong Province Statistical Yearbook(tjj.shandong.gov.cn (accessed on 18 May 2022)); (6) soil data are collected and obtained from the World Soil Database (HWSD) (http://westdc.westgis.ac.cn/ (accessed on 6 November 2021)); (7) drawing on existing landscape studies [40,41,42], this paper uses 30 m resolution land-use data selected to calculate landscape shape index (LSI), contagion (CON), Shannon’s diversity index (SHDI), and patch density (PD) to characterize the landscape pattern of SYRB. The four dimensions of density, diversity, shape and aggregation were measured and analyzed.

### 2.2. Study Area

The Yellow River, known as China’s “Mother River,” is the world’s fifth largest river and the second largest in China. It is the principal cradle of Chinese civilization. It originates on the Qinghai–Tibet Plateau and flows through nine provinces before joining the Bohai Sea at Dongying city, Shandong Province, with a total length of 5464 km. The study area of this paper is the Shandong part of the Yellow River (34°58′−38°09′ N, 114°48′−119°19′ E), with a total length of 628 km, accounting for 11.5% of the total length of the Yellow River, and a basin area of 18,300 km^2^, mainly covering the eastern and central parts of the topography of the nine cities of Heze, Jining, Liaocheng, Dezhou, Jinan, Tai’an, Zibo, Binzhou and Dongying. The terrain of the study area is high in the east and middle, and is a low mountain and hilly area. The vast area in the north and south is relatively flat, and is a fertile plain area. It is the main grain production area and the main agricultural water demand area in Shandong Province (Figure 1). The climate is a warm–temperate monsoon with four distinct seasons, 500–800 mm of annual precipitation, and 12–14 °C of average annual temperature. The study area has witnessed significant urbanization since 2000, with the share of built-up land area increasing from 13.14% in 2000 to 18.13% in 2020, and city expansion has taken up a large quantity of arable land, forest land, and other land types. The structure of the ecosystem has changed as a result of the rapid shift of land-use types, and its functions have been considerably impacted [43].

### 2.3. Methods

#### 2.3.1. Evaluation of Ecosystem Service

The ESs were selected based on the following principles, taking into account the policy background, development status and previous studies on ecosystem services: Firstly, the classification indicators of the Millennium Ecosystem Assessment (MA) ensured comparability between this study and previous studies. Secondly, the ESs that reflect the characteristics of the study area were selected based on previous studies in the Yellow River Basin and other representative areas. Finally, the ESs were selected based on the accessibility and accuracy of the required data [44]. In the end, four key ecosystem services were selected as research objects, namely Habitat Quality (HQ), Carbon Storage (CS), Water Yield (WY) and Food Production (FP).

Although there are many types of ecosystem services, these four ecosystem services are typical and important types of ecosystem services in the Shandong Yellow River Basin. Each of these services has a significant impact on social life. Firstly, food is the basis for economic development, social stability and national independence [45]. The Shandong Yellow River Basin (SYRB) is an important grain producing area in Shandong. In 2020, the total grain production in the region accounted for 70% of Shandong, but the area only accounted for 50% of Shandong, which indicates a strong grain production capacity [46]. Hence, exploring the SYRB Food Production services (FP) can help further explore the advantages of regional grain production resources in a major grain-producing province. Secondly, due to the rapid development of China’s economy and the increase of urbanization level, a slew of issues such as water scarcity, water environment deterioration, and severe water pollution have risen [47], and the Water Yield (WY) services of ecosystems have gained more attention [48]. Shandong’s Yellow River Basin is rich in water resources. The region has developed industries and agriculture that consume large amounts of water resources [49]. As ecosystems’ water yield services not only offer water resources for diverse internal ecological groups, but also continue to meet the consumption of external water resources, so they occupy a very important position in many ecological service functions [50]. Therefore, the evaluation of water-producing services is an essential element in ecological zoning at different scales. Thirdly, to some extent, Habitat Quality (HQ) can characterize a region’s biodiversity. On the one hand, biodiversity is a guarantee for human society’s progress and is critical for agriculture’s long-term development. On the other hand, biodiversity is certainly the foundation for the sustainable growth of other ecological services and a key indicator for assessing ecosystem health. Therefore, it is critical to investigate the level of HQ in SYRB in order to evaluate its ecological service functions [51]. Fourthly, terrestrial Carbon Storage (CS) plays an essential role in global carbon sinks. Carbon fixation plays an important role in climate regulation in ecosystem services and is an important indicator to measure ecosystem functions in a region. Hence, boosting regional ecosystem carbon stocks can effectively lower CO_2_ levels in the atmosphere, increase the carbon sequestration function of terrestrial ecosystems, and ameliorate climate change [52].

The CS studied in this paper is mainly divided into the aboveground part of carbon storage and the underground part of carbon storage. The aboveground part of carbon storage of CS is derived from the amount of net primary productivity, while the underground part of carbon storage focuses on soil organic carbon [52,53]. Using the Carnegie–Ames–Stanford approach (CASA), the Integrated Valuation of Ecosystem Services and Trade-offs (InVEST), and the Food Production Estimation Model, this paper utilizes GIS technology to quantify ESs, and reveals the spatial distribution characteristics of ecosystem service relationships in 2000, 2010 and 2020. This paper refers to previous studies to calculate the four ESs [25,54,55,56].The processes that were used to assess the ESs in the study are listed in Table 1.

#### 2.3.2. Calculation of Trade-Offs

The root mean square deviation (RMSD) is a simple yet effective method for calculating the degree of trade-off between any two or more ecosystem services [57]. RMSD describes the dispersion from the mean ES standard deviation by measuring the difference between the individual ES standard deviation and the mean ES standard deviation, and it extends the meaning of trade-off from a negative correlation to the rate of non-uniformity of change between ESs in the same direction. This enables a more detailed portrayal of the degree of interaction between ESs. This research quantifies the link between ESs using RMSD, which characterizes the trade-offs of ecosystem services by the distance from a pair of ecosystem service coordinates to a 1:1 line, with the relative position of the coordinates representing the relative benefit of a certain ES (Figure 2). The greater the RMSD value is, the stronger the trade-off intensity is. On the contrary, the smaller the RMSD value is, the weaker the trade-off intensity is, and the two ecosystem services tend to synergetic. When the RMSD value is zero, the two ESs have a synergistic relationship. It should be noted that data must be normalized to remove the influence of magnitude before calculating the RMSD. The ES standardization is defined as [57]:(1)$$\;ES_{std}=\frac{\left(ES_{obs}-ES_{min}\right)}{\left(ES_{max}-ES_{min}\right)},$$
where $\;ES_{std}$ is the standardized ES, representing the relative income of the ES, and its value ranges from 0 to 1. $ES_{obs}$ is the observed value of ES; $ES_{min}$ and $ES_{max}$ are the minimum and maximum observed values of ES, respectively.

In Figure 2, points A, B, C, and D are the coordinate points of a specific pair of ESs under different conditions, and RMSD is the distance between the coordinate point and the 1:1 line. The trade-off for ecosystem services is 0 when the coordinate point is on the 1:1 line, and the further the coordinate point deviates from the 1:1 line (i.e., where the arrow points), the greater the trade-off of ecosystem services. In Figure 2, the order of trade-off of each point is C = D > A > B, but point C implies a bias toward ecosystem service 1, whereas point D shows a bias toward ecosystem service 2.

On the basis of the standardization of ESs, the RMSD values are calculated as follows.
(2)$$\mathrm{RMSD}=\sqrt{\frac{1}{n-1}\times \overset{n}{\underset{i=1}{\sum }}{\left(ES_{i}-\bar{ES}\right)}^{2}}$$
where $ES_{i}$ is the standard value of the *i*th and $\bar{ES}$ is the average of the two ESs involved in the calculation.

#### 2.3.3. Measurement of Landscape Pattern Index

The spatial form of cities is as important as the influence of socio-economic and natural factors on the ecological environment [58]. This paper measures and analyzes the spatial forms of regions in terms of density (LSI), diversity (CONTAG), shape (SHDI) and agglomeration (PD) (Table 2).

#### 2.3.4. Discrimination of Driving Factors

Based on the research on trade-off analyses between ESs and the SYRB’s socioeconomic and natural environmental context [59,60], we chose indicators from natural factors [61,62], land-use factors [63,64,65], landscape configuration factors [66,67] and socio-economic factors [68,69], and refer to previous studies on the driving factors of ESs; a system of assessment indicators for the elements influencing trade-offs between ESs was developed (Table 3).

ALM is a powerful technique for developing multivariate models and selecting independent variables. It has two advantages. First, the model can include as many independent variables as possible, especially the variables that have a significant impact on the dependent variables; second, in the model results of stepwise regression, there are no independent variables that have no significant influence on the dependent variables, and the respective variables do not have serious multicollinearity [70].

The basic idea of ALM is to introduce all the independent variables that have significant effects on the dependent variable into the regression model, so that the final regression model is optimal. Firstly, the variables are introduced into the regression model one by one in a given order, and the significance of each variable is tested. Secondly, when the variables introduced later in the model render the variables already included in the model insignificant, the previously introduced independent variables are eliminated to ensure that all variables in the model are significant. Finally, an F-test is performed on the regression model before introducing new independent variables at each step, until no significant independent variables are introduced into the regression model and no insignificant independent variables are excluded from the regression model.

In this study, the comprehensive landscape pattern indexes, several natural and socio-economic factors were taken as independent variables, and RMSD was taken as dependent variable. Stepwise linear regression was used to explore the main driving factors of RMSD in each city. Standardized regression coefficients were also utilized to examine the impact of each component on RMSD and to identify the variability and strength of major drivers’ contributions across cities.

Suppose there are n variables $x_{1}$, $x_{2}$, ⋯, $x_{n}$. The regression model using n independent variables is called the full model and its basic formula is as follows:(3)$$y=\beta_{0}+\beta_{1}x_{1}+\beta_{2}x_{2}+\cdots +\beta_{n}x_{n}+\varepsilon$$

When $x_{i}$ is removed from the n independent variables, the regression model for which there are (n−1) independent variables at this point is called the constructed model.
(4)$$y_{i}=\beta_{0}+\beta_{1}x_{1}+\beta_{2}x_{2}+\cdots +\beta_{i-1}x_{i-1}+\beta_{i+1}x_{i+1}+\cdots +\beta_{n}x_{n}+\varepsilon$$
(5)$$\Delta r_{i}^{2}=r^{2}-r_{i}^{2}$$
where $r^{2}$ is the correlation coefficient of the full model; $r_{i}^{2}$ is the correlation coefficient of the constructed model. When $\Delta r_{i}^{2}$ is not significantly zero, i.e., that is, the variables $x_{1}+x_{2}+\cdots +x_{i-1}+x_{i+1}+\cdots +x_{n}$ are already present in the regression model, the introduction of the independent variable $x_{i}$ will significantly improve the explanatory power of the dependent variable y. Otherwise, when $\Delta r_{i}^{2}$ is significantly zero, the removal of the independent variable $x_{i}$ has no significant effect on the explanatory power of the dependent variable y  in the full model.

#### 2.3.5. Analysis of Influence Degree of Each Driving Factor on Different Research Units

The GWR model was used in this study to express the degree of influence of each driver on distinct study units [71]. Because the ALM model eliminates the problem of variable multicollinearity, the important influencing factors identified by ALM can be used as explanatory variables in the GWR model. The GWR is a regression model that differs from the classic global regression model, and it is generated by expanding the traditional regression model by taking spatially non-smoothness into account. The regression model has an advantage over the typical global regression model in that each study unit has its own regression coefficient value, which can better reflect the degree of influence of each driver on multiple research units and capture the spatial variation of the drivers [72]. The GWR analysis was conducted in the MGWR 2.2 tool with the following equation:(6)$$Y_{i}=\beta_{0}\left(u_{i},v_{i}\right)+\overset{i}{\underset{k=1}{\sum }}\beta_{k}\left(u_{i},v_{i}\right)X_{ik}+\tau ,$$
where $Y_{i}$ is the RMSD of sub-basin i; $X_{ik}$ is the value of the kth explanatory variable at sub-basin i; $\left(u_{i},v_{i}\right)$ is the coordinate of basin i; $\beta_{0}\left(u_{i},v_{i}\right)$ is the spatial intercept at sub-basin i; $\beta_{k}\left(u_{i},v_{i}\right)$ is the regression coefficient of the *k*th explanatory variable at basin  i, locally estimated using weighted least squares; τ is the residual.

## 3. Results

### 3.1. ESs Variability in the Shandong Yellow River Basin from 2000 to 2020

This paper assessed the availability of HQ, CS, WY, and FP services in the SYRB in 2000, 2010, and 2020 with a spatial resolution of 1 km (Figure 3).

CS has changed in spatial distribution pattern and increased in time. Temporally, the mean values of CS in 2000, 2010 and 2020 are 1639.32 t, 1713.78 t and 1764.42 t, respectively, showing an increasing trend. Spatially, it mainly shows an increase in the north and a decrease in the south. The increase of CS in the north of SYRB is due to the protection of the Yellow River Delta area, which plays a major role in the improvement of carbon reserves in this area. However, CS in the southern part of the SYRB has decreased, which is mainly due to the decrease in the area of cultivated land and the intensification of human activities. The central part of SYRB has abundant precipitation, and the land-use type is basically woodland, with high vegetation coverage and high organic matter content in the soil [73]. Therefore, CS in this area has been at a relatively high level in the past 20 years.

FP is basically stable in the spatial pattern and slightly increased over time. The three periods’ average FP values are 4.21 t, 4.33 t, and 4.51 t, respectively. FP exhibited “high in the southwest and low in the northeast” distribution characteristics. This is mainly due to the influence of soil salinization in the northeastern part, which is not suitable for planting food crops. However, the soil is fertile and flat in the southwest of the study area, and the land-use type is mostly cultivated land, which has always been a typical agricultural area, so the FP in this area is relatively high. In addition, with the progress of science and technology, FP has maintained a slight growth trend in time.

WY fluctuates over time and spatially varies significantly, but the spatial distribution pattern is relatively stable. WY fluctuates greatly in time, with average values of 178.87 mm, 355.16 mm and 325.12 mm for the three periods. Meanwhile, WY shows a spatial distribution pattern of “high in the center and low in the surroundings”, with the high values mainly concentrated in Tai’an and the southern part of Jinan, which are mostly mountainous and hilly areas with lush vegetation distribution. This area has a strong water storage and retention capacity and abundant precipitation, so it has a high water production capacity. Similarly, the southern part of the study area also has a high water yield, with the largest freshwater lake in the north and abundant water resources. By contrast, Binzhou and the northern part of Dezhou have a lower water yield than the other areas, as they are mostly plain areas with high population densities and concentrated arable land, making them less productive [74].

HQ varies significantly in space and little in time. HQ varies less over time, from 0.639 in 2000 to 0.637 in 2010 and 0.633 in 2020. The spatial distribution of HQ is relatively consistent with that of WY, showing a pattern of low in the surrounding area and high in the middle. The central region is less disturbed by human activities and has fewer threat sources, so the biodiversity level is high. However, cultivated land, as one of the threat sources, is highly disturbed by human activities and affects the quality of nearby habitats, resulting in a lower HQ [75]. In general, the distribution of HQ low-value areas is highly consistent with that of land types with high interference from human activities, especially construction land, which is mainly concentrated in towns, villages and other construction areas.

### 3.2. Quantification of Trade-Offs between ESs

The RMSD index was used to measure the trade-off intensity and perform statistical analysis on the value range (Figure 4), and to explore the spatial heterogeneity of trade-off relationships (Figure 5).

As a whole (Figure 4), the trade-off strength of FP_CS was low, with a 20-year mean of 0.071. The trade-off strength of FP_WY (0.319), WY_CS (0.323) and WY_HQ (0.315) fluctuated around 0.32. CS_HQ has the highest trade-off strength of 0.505, followed by FP_HQ (0.429).

WY-FP trade-off showed little spatial variation over the 20-year period (Figure 5), with high trade-off (RMSD > 0.5) areas concentrated in the north-western part of the SYRB. On the contrary, low trade-offs (RMSD < 0.1) are mainly concentrated in the eastern mountainous areas. The intensity of the trade-off is highest in 2000, followed by 2020, and the overall trade-off show a trend of decreasing before increasing.

The spatial variation of WY-HQ trade-off over the 20-year period is obvious (Figure 5). In 2000, high trade-off areas were primarily concentrated in the northwestern part of the SYRB, such as Wudi, Yangxin, Lingxian and Linyi, while the low trade-off areas were mainly concentrated in the central and eastern regions, including Yiyuan, Zouping and other regions. In 2010, the high-weight areas shifted to the south, and the low-weight areas became more dispersed. In 2020, the high trade-off areas changed again, concentrated in the west, and the low trade-off areas changed to the East. Over time, the overall intensity of the trade-off continues to decline.

The spatial distribution of the WY-SC trade-off is almost consistent with that of WY-FP (Figure 5). In 2000, high trade-off areas were mainly concentrated in the northwest of SYRB, while low trade-off areas were mainly concentrated in the eastern mountains. The change in time of WY-SC trade-off is also similar to that of WY-FP, showing a downward trend first and then an upward trend. Among them, the trade-off intensity is the highest in 2020, followed by 2000.

There is variation in the spatial distribution of high trade-offs in HQ-FP (Figure 5), while there is little variation in low trade-off areas. There is a gradual shift from south to north, with a decreasing number of high-weight areas in the south and an increasing number in the north. Low-weight areas show little change and are mostly located in the eastern part of the SYRB. The intensity of trade-offs was highest in 2010, followed by 2000. Overall, the intensity of the trade-off rises first and then decreases.

The spatial distribution of high trade-offs in HQ-CS has been concentrated in the south, while there is some variation in the distribution of low trade-off areas (Figure 5). High-weighted areas are mostly concentrated in the southern part of the study area, including Chengwu, Cao County, Shan County and Dingtao. The distribution of low-equilibrium areas varies between years, with fewer high-equilibrium areas in 2000, including Feicheng and Pingyin in the central region and Qingyun and Wudi in the northern region, and fewer high-equilibrium areas in 2010, including Kenli and Lijin, as well as Leling and Linyi. In 2020, the number of high trade-off areas increased and mainly concentrated in the central region, including Zhangqiu, Zouping, Hengtai, Laiwu and other regions. In general, the intensity of the trade-off rises first and then decreases. The intensity of the trade-off was highest in 2010, followed by 2020.

The spatial distribution of the FP-CS high trade-off shifts from north to south, while the low trade-off shifts from the central south to the central north (Figure 5). In 2000 and 2010, the low-weight areas were concentrated in the central and southern parts of the study area, while in 2020 the low-weight areas were concentrated in the northern and central parts. On the whole, the intensity of the trade-off rises first and then decreases. The intensity of the trade-off was highest in 2010, followed by 2020.

We summed the six trade-off intensities to obtain the overall trade-off intensity for each county and calculated its trend (Figure 6).

The intensity of the trade-offs in SYRB changed little in the spatial patterns between two decades. Spatially, the overall trade-off intensity is higher in the western, northwestern, and southwestern regions, and lower in the central–eastern and northern northeastern regions. Temporally, the average value of the overall trade-off was 2.15 in 2000, 2.10 in 2010, and 2.21 in 2020, respectively.

The trend of change from 2000 to 2010 is opposite to the trend of change from 2010 to 2020. From 2000 to 2010, the regions where the total trade-off intensity decreased are concentrated in the mid-west, and the other regions are those where the trade-off intensity increases; from 2010 to 2020, the regions where the total trade-off intensity increased are concentrated in the central and western regions, and the other regions are the regions where the trade-off intensity decreases.

### 3.3. Relative Benefits of ES in the Trade-Offs for Different Regions

The relative benefits of ecosystem service trade-offs tend to differ across locations in different time periods (Figure 7). Among them:

In the trade-off between WY and FP, the number of areas with higher FP benefits is much larger than the number of areas with higher WY benefits, and the distribution pattern is more stable; The areas with higher WY benefits are concentrated at the mouth of the Yellow River and in the mountains of Jinan, while the areas with higher FP benefits are mostly in the plains, where there is a large amount of contiguous arable land, which allows FP to obtain higher benefits in the trade-off.

In the trade-off between WY and HQ, the number of areas with higher WY returns is slightly larger than the number of areas with higher HQ returns, and the spatial distribution shows a north–south difference, with higher HQ returns in the south and higher WY returns in the north in 2000 and 2010; however, this distribution pattern changed to some extent in 2020. The areas with higher HQ returns are scattered in the southern and northern parts of the study area as well as sporadically in the central part, while the areas with higher WY returns are mainly located in the central part of the study area in 2020. The protection of the Yellow River estuary plays a large role in the improvement of HQ in Dongying, while the return of farmland to forestry allowed a large improvement of HQ in the central part of the study area.

In the trade-off between WY and CS, the number of areas with higher CS gains is much larger than the number of areas with higher WY gains, and the spatial distribution pattern is more stable. The areas with higher WY gains are mostly located in the north, and some areas in the south also had higher WY gains in 2010 and 2020. However, the areas with higher gains in CS in this trade-off are somewhat similar to those with higher gains in FP in the WY_FP trade-off, because arable land is more widely distributed in the study area, and arable land is the main land use type for carbon storage.

In the trade-off between HQ and FP, the number of areas with higher FP yield is much larger than the number of areas with higher HQ yield, and the spatial distribution pattern is more stable, with areas with higher HQ yield scattered in the north and east, mainly in the Yellow River delta and mountainous areas, but the number of these areas has decreased over the past 20 years. This has kept their HQ at a relatively high level.

In the trade-off between HQ and CS, the number of areas with higher HQ gains is slightly greater than the number of areas with higher CS gains, and the spatial pattern has always been a stable difference between east and west, with areas with higher CS gains concentrated in the central and western sides.

From the perspective of time change, there are differences in the number of regions with different ecosystem service trade-offs with relative benefits. In terms of time, in 2000, there were 63 regions with higher WY returns, 73 regions with higher HQ returns, 113 regions with higher CS returns, and 147 regions with higher FP returns; In 2010, there were 82 regions with higher WY returns, 67 regions with higher HQ returns, 103 regions with higher CS returns, and 144 regions with higher FP returns; In 2020, there were 75 areas with higher WY gains, 68 areas with higher HQ gains, 97 areas with higher CS gains, and 156 areas with higher FP gains.

In SYRB, FP has an absolute advantage in the trade-offs, followed by CS, while HQ has the least number of regions with an advantage in the trade-offs. The number of areas with higher FP gains in the three time periods is much larger than the other three services, because SYRB is the main grain-producing area in Shandong, arable land has an absolute advantage in the land use type, and the government also attaches higher importance to FP than other ecosystem services, so it is in an advantageous position in the trade-offs with other services. The government attaches less importance to HQ than other services, and policy adjustment will give priority to sacrificing HQ for other ecological services, which makes HQ in a disadvantageous position in the trade-offs.

We counted each service in each region in Figure 7 that had a higher relative gain in the trade-offs (Figure 8). When a service has the largest number of times in the summary of relative benefits of various tradeoffs in a region, we considered that service to be the prominent ecosystem service in that region; when a region did not have a service that was dominant in the trade-off the most, we considered that region to have no prominent ecosystem service.

In Figure 8, areas with prominent FP are the most numerous and mainly concentrated in the west, northwestern southwest and other regions, while areas with prominent CS are the least numerous, with only three regions in 2000, developing to nine in 2010 and decreasing to two in 2020. In addition, areas with prominent WY, mainly concentrated in the northeast, had a decreasing trend in number. Areas with prominent HQ, with a large spatial variation, were mainly concentrated in the central–eastern region in 2000, decreasing in number and more sporadically distributed in 2010, and increasing in number in 2020, mainly concentrated in the northern region. Areas without prominent ESs are spatially concentrated in the central and eastern regions, with an increasing and then decreasing trend in number.

### 3.4. Identification of Key Drivers for the Trade-Off among ESs

The drivers of the trade-offs between ESs can be better explained using the ALM model, and the results show that natural factors, social factors, and landscape indices all influence trade-offs between ESs to varying degrees (Table 4, Table 5 and Table 6).

CLP and NDVI are the main factors positively affecting the trade-off relationship between FP_HQ. The contribution of CLP was 37.2%, 24.5%, and 31.3% in 2000, 2010, and 2020 respectively, and the shadow contribution of NDVI to the trade-off relationship between FP_HQ was 18.3%, 37.5%, and 31.3% respectively. In this trade-off, the influence nature of each influencing factor is either positive or negative. The positive influence factors decrease while the negative influence factors increase. There were four positive influences and two negative factors in 2000, five positive influences and two negative influences in 2010, and three positive factors and four negative factors in 2020.

The main factors affecting the trade-off between FP_CS have changed, from cultivated land area to NDVI, which is mainly negatively influenced by the proportion of arable land area in 2000 and 2010, with a contribution of up to 29.7% and 31.1% respectively, while in 2020, NDVI had the greatest influence (31.6%) and is negative. In this trade-off, the number of influencing factors that have a positive impact and a negative impact is different. There were two positive influences and seven negative factors, respectively, in 2000 and 2010, and three positive factors and four negative factors in 2020.

The main factor influencing the trade-off between FP_WY has changed from PRE to NDVI. PRE had a significant influence in 2000 (53.4%) and 2010 (35.1%), both of which were negative, while in 2020 the trade-off between FP_WY was mainly affected by the positive influence of NDVI (24.1%) and the negative influence of TEM (16.6%). In this trade-off, the number of influencing factors that have a positive impact or a negative impact changed. There were six positive influences and two negative factors in 2000, three positive influences and four negative influences in 2010, and five positive factors and two negative factors in 2020.

The main factor influencing the trade-off between CS_HQ changed from NDVI to proportion of woodland area. The negative impact of the proportion of forest land area has gradually increased, from 16.2% in 2000 to 51.1% in 2010 and 37.1% in 2020, while the negative impact of NDVI has greatly decreased, from 31.4% in 2000 to 9.2% in 2010 and 4.2% in 2020. In this trade-off, the number of influencing factors that play a negative role increased. There were four positive influences and three negative factors in 2000, three positive influences and five negative influences in 2010, and four positive factors and five negative factors in 2020.

The main factor influencing the trade-off between WY_HQ has changed from CON to PRE, and then to CON. In 2010, PRE had the greatest active impact on WY_HQ, accounting for 61.3%. In 2000 and 2020, CON had the largest impact on WY_HQ, and both values were positive, 38.1% and 31.2%, respectively. In this trade-off, both positive and negative factors increased. There were four positive influences and one negative influence in 2000, three positive influences and five negative influences in 2010, and five positive influences and four negative influences in 2020.

CON is the main factor influencing the trade-off relationship between WY_CS, with contributions of 50.7%, 32.6%, and 41.5% respectively. Secondly, TEM and PRE also have a certain impact on the trade-off relationship. In 2010, TEM had a significant negative impact (41.6%), and in 2020, PRE had a significant negative impact (31.2%), while the remaining factors had a weak or no impact. In this trade-off, the overall performance is that the factors that play a negative role increased. There were two positive influences and four negative factors in 2000, three positive influences and three negative influences in 2010, and two positive factors and six negative factors in 2020.

### 3.5. Spatial Heterogeneity Analysis of ESs Trade-Off Drivers

Spatially, the R^2^ between the six pairs of ESs exceeds 0.50 in most areas (Figure 9), indicating that the GWR model could better explain the influence of each factor on the heterogeneity of the spatial distribution of the trade-off relationship between the pairs of ESs in SYRB.

For the FP_HQ trade-off (Figure 10), the dominant factor is NDVI in the largest area in all three periods, accounting for 65%, 68%, and 59% of the total area, respectively, and is mainly concentrated in the south and west, both of which are negatively influenced. The dominant factors in the other regions varied, with TEM having a greater impact on the north-eastern part of the SYRB in 2000, with both positive and negative impacts, accounting for 29% of the total area. In 2010, the dominant factors in different areas of the northern part of the SYRB included TEM (2%), CLP (8%), GDP (6%), and COM (6%). In 2020, in addition to the areas affected by NDVI, some areas in the north were greatly affected by CON (21%), TEM (11%) and PRE (9%).

For the FP_CS trade-off (Figure 11), CLP (15%) and TEM (20%) had a greater influence on the FP_CS trade-off in the central region during 2000, and both are positive. In the southern part and the northern region, arable land area (24%), NDVI (21%), and PD (20%) play a dominant role. Moreover, in both 2010 and 2020, NDVI was the dominant factor with the largest area, 59%, and 30% respectively. The difference is that in 2010 the main influence was concentrated in the central and southern parts of the SYRB and is positive, while in 2020 it was concentrated in the central part and is negative.

For the FP_WY trade-off (Figure 12), the largest area was dominated by PRE in 2000, 2010, and 2020, with 83%, 30%, and 24% of the total area, respectively, and all of them are negatively influenced. In 2010, NDVI (23%), TEM (17%), CON (17%) and GDP (14%), also played a dominant role in the FP_WY trade-off relationship in different areas. In 2020, in addition to PRE, NDVI (24%) and the proportion of cultivated area (23%) also had a strong influence.

For the trade-off relationship between CS_HQ (Figure 13), the area with NDVI as the dominant factor is the largest, which was 48% in 2000, 64% in 2010, and 52% in 2020. Besides the coastal area in the north of SYRB in 2020, all other areas are positively affected. However, the regions dominated by CON (23%) and POP (12%) and most regions dominated by LSI (17%) are negatively affected. In addition, in 2010, the proportion of grassland area also played a negative leading role in 29% of the regions, and 8% of the regions were greatly affected by CON. In 2020, POP (30%) positively dominated the trade-off between CS_HQ in the central region and CON (18%) negatively dominated in the southern region.

In the WY_HQ trade-off (Figure 14), the three-period dominant factors are CON with the largest area, accounting for 73%, 38%, and 59% of the total SYRB area respectively, and all are positively dominant factors. In 2000, the counties with CON as the leading factor were located in the western and southern regions, while those with NDVI (21%) as the negative leading factor were concentrated in the eastern region. In 2010, the counties with CON as the dominant factor were distributed in the eastern part of the SYBR, while the areas with NDVI as the positive dominant factor were located in the western part of the SYRB. In 2020, there were east–west differences in the main drivers of trade-offs, with PRE as the negative dominant driver on the west side with 42% of the area, and CON as the positive dominant driver on the east side. NDVI (8%) is mainly distributed in scattered areas, some of which are positively affected, and some are negatively affected.

For the CS_HQ trade-off (Figure 15), CON has the largest impact, with 100% in 2000, 59% in 2010, and 35% in 2020, all with positive impacts. In 2010, the CON-dominated areas were concentrated in the western and southern parts of the SYRB, while the areas dominated by TEM (17%) were located in the east and CLP (14%) and arable land areas (11%) were located in the north. In 2020, in addition to CON as the dominant factor in the eastern region, TEM (26%), PRE (11%), and CLP (29%) also played a dominant role in the southern, northern, and western regions of the SYRB, respectively.

## 4. Discussion

### 4.1. Accuracy of Ecosystem Services Evaluation

This paper analyzed the accuracy of each model assessment result by comparing the measured data with previous studies to further ensure the quality and accuracy of the ecosystem service assessment results. Considering that a comprehensive simulation study on ESs has not been carried out in this study area, and there are few measured data, we have compared and analyzed the evaluation results with the relevant research results of SYRB by others.

The annual average values of WY in the study area in 2000, 2010, and 2020 were 178.97 mm, 355.16 mm, and 325.12 mm, respectively, which were slightly less than those published in the Shandong Province Water Resources Bulletin, being 184 mm, 376 mm, and 344 mm, respectively. Similarly, the average FP calculated in this paper from 2000 to 2020 is 4.68 t, which is only 0.23 t smaller than the average FP calculated by Geng in SYRB from 2000 to 2018, which is 4.91 t [76]. The difference is due to the use of NDVI data from different time periods. However, the evaluation results of CS and HQ are more consistent with the research results of others. The accuracy of CS is primarily related to NPP [77], and the accuracy of CS is determined by comparing NPP values. The CS evaluation result in this paper agrees with Yang’s findings [78]. Song et al. calculated HQs of 0.642 and 0.637 in 2000 and 2018 [79], respectively, which are comparable to 0.639 in 2000 and 0.637 in 2020 in this paper. The HQ deviation is primarily due to different weights being assigned to different threat sources. In conclusion, there is a reasonable margin of error between the evaluation results of this paper and previous studies, and the assessment results of each ES are reliable.

### 4.2. Analysis of Trade-Off Intensity and Relative Benefits between ESs

At the county scale, there is spatial heterogeneity in the spatial distribution of trade-off intensities among different ESs, but the overall spatial distribution of trade-off intensities shows higher trade-off intensities in the western, northwestern, and southwestern regions, while the overall trade-off intensities are lower in the central–eastern and northeastern regions.

In the trade-off relationship, the distribution of areas with prominent FP has similarity with the distribution of areas with higher total weighting. That is, FP is more dominant in the trade-offs in areas with higher trade-off strength. However, areas without prominent ESs are mostly concentrated in areas where the trade-off intensity is low.

In areas where the trade-off intensity is high, arable land is widely distributed and agriculture is the main source of income for farmers in the area. In order to obtain higher income, farmers have converted a large amount of forest and grassland to cropland [46], which puts FP in an advantageous position in the trade-off with other ESs. At the same time, the reduction of forest and grassland has increased the conflict between different ecosystem services [80], which also puts the region in a relatively high trade-off intensity.

In the areas where the trade-off intensity is low, mostly located in the mountainous areas of Taishan, woodlands and grasslands are widely distributed. The protection of forests has been an important policy of the local government, and the area is less disturbed by human activities, which makes the conflict between regional ESs relatively moderate, and thus the area is at a low level of trade-off intensity. This has led to a low level of trade-off between ESs, and because there is little human interference, humans rarely adjust ESs to their own needs in order to obtain higher levels of desired services, resulting in the absence of prominent ESs in the region.

### 4.3. Explanation of the Main Drivers of Different Ecosystem Service Trade-Offs

Since the results of 2020 reflect the impact mechanism of ES trade-offs in recent years, it will be more reasonable to analyze the results of 2020 and put forward corresponding suggestions on ES regulation [5]. In addition, considering the length, importance and future application of the article, this paper only discusses the results for 2020.

The driving factors of ES trade-offs mainly include NDVI, TEM, forest land area proportion, cultivated land area proportion, CLP, CON, SHDI and PD, while the contribution of other influencing factors is low (CR < 10%). Among the drivers of ES trade-offs, TEM and PRE are heavily influenced by local geographic conditions, which are difficult or impossible to change in a short period. In contrast, other natural and social factors are the easiest for policymakers and local residents to adjust in order to achieve higher desired ESs [81].

NDVI promotes CS and HQ more obviously and FP weakly [8,9]. Therefore, NDVI promotes FP_ HQ trade-offs and FP_ CS trade-offs. This is mainly because in areas with high NDVI, there are abundant plants, and the growth of plants will increase the amount of carbon fixation. At the same time, the landscape connectivity of these areas is better [7], and the habitat quality is higher. In addition, plant evapotranspiration is higher and water production is lower in high-NDVI areas [16]. As a result, NDVI also improves the FP_WY trade-offs.

Compared to other land use types, on the one hand, woodland is less disturbed by human activities, and there are fewer other sources of threats to the forest land, so the carbon sequestration rate will be higher [14]. On the other hand, with the increase of forest land area, the area of arable land will be relatively reduced, and the grain production it provides will also be reduced. Moreover, the transpiration of plants in the forest land will reduce water production, enhance CS and weaken FP. Therefore, the forest land area plays a role in promoting the FP_HQ trade-off, FP_CS trade-off and WY_CS trade-off, while the forest land area has a negative impact on the CS_HQ trade-off, FP_WY trade-off and WY_CS trade-off. In particular, the forest land area has a great impact on the CS_HQ trade-off.

The impact of the proportion of cultivated land area on the FP_CS trade-off is particularly prominent. Different from the proportion of forest land area, the proportion of cultivated land area realizes the purpose of promoting trade-off by strengthening FP and weakening CS. As the cultivated land area increases, FP will increase accordingly. Therefore, the proportion of cultivated land area has a certain impact on the FP_HQ trade-off and the FP_WY trade-off.

CLP can reflect the situation of urban development. With the rise of CLP, a large number of ecological lands has been converted into construction lands [11]. On the one hand, this transformation results in the reduction of vegetation coverage; on the other hand, it also results in the high fragmentation of the landscape, which leads to the deterioration of CS and HQ [12,13]. As a result, CLP promotes the FP _HQ and FP_ CS trade-offs [56].

A higher CON value indicates that a specific plate is well connected. In the SYRB area, due to the influence of natural factors, social factors, economic factors, systems and land contract responsibility system policies, the problem of farmland fragmentation is prominent. Hence, a larger CON indicates less arable land and less human activity disturbance. However, a higher CON increases the subsurface carbon content in the CS, and a lower CON decreases the HQ [82]. As a result, CON has some influence over the WY_HQ trade-off, the FP_CS trade-off, the FP_WY trade-off, and the WY_CS trade-off.

SHDI and PD can reflect the degree of damage to the parcel by human activities from different perspectives, with PD showing heterogeneity per unit area and SHDI emphasizing the contribution of rare land types to the degree of fragmentation [83]. Larger SHDI and PD indicate stronger human disturbance, which results in lower above-ground carbon content in HQ and CS [26], thus having an impact on the trade-off between CS_HQ.

### 4.4. Policy Implication

Although precipitation and temperature have a significant impact on the intensity of the SYRB ecosystem services’ trade-offs, socio-economic development has also had a significant impact on the evolution of the trade-off relationship. For example, changes in landscape patterns and a decrease in NDVI have resulted from increased urbanization [84]. Meanwhile, population and GDP growth, as well as landscape fragmentation, have all had an impact on ecosystem services [84]. Therefore, a trade-off analysis of ecosystem services can effectively balance regional economic development and environmental protection. The findings suggest that there is a conflict between the ecosystem services provided by agricultural land and ecological land (woodland and grassland), as well as a competitive relationship between food production and ecological conservation [85].

Some eco-environmental protection strategies and actions will improve the regulation of ecosystem services such as air purification, carbon sequestration and oxygen release, and soil and water conservation quickly, but may reduce food production [86]. However, SYRB is a significant food-producing region. Therefore, from the standpoint of food security, there is a need to focus on the trade-offs between food production and other ecosystem services while improving the ecological environment’s quality. In this area, future planning should make a reasonable allocation of agricultural land and ecological land, and the three control lines of “ecological protection red line” “permanent basic agricultural land” and “urban development boundary” should be scientifically and reasonably defined in the country’s spatial planning, so as to achieve a win-win situation in grain production, ecological protection and socio-economic development [87]. On the one hand, food cultivation should be guaranteed in high-quality arable land areas such as Binzhou, Dezhou, and other cities, and farmers’ motivation to grow food should be enhanced. At the same time, modern agricultural technology and advanced agricultural facilities should be used to increase food production, and regional resource advantages should be brought into play to ensure food supply. On the other hand, in the mountainous areas of Tai’an and Jinan, we should reasonably promote the return of farmland to forest, improve the quality of the regional ecological environment, and strengthen the regulating service function by combining land suitability analysis. With China’s carbon peaking and carbon neutral targets, it is necessary not only to control the expansion of construction land and maintain a certain proportion of ecological land, but also to develop low carbon industrial methods, such as the use of renewable energy, improve energy efficiency, and reduce carbon emission intensity, in order to balance carbon storage and emissions, particularly in areas such as Zibo and Laiwu where secondary industries are more developed.

The site is rich in water resources and has the largest freshwater lake in northern China. It is especially important in water purification and quality protection. Furthermore, ecological corridor construction projects within the region’s cities should be continued, and greenways along the Yellow River and the highway belt should be developed. Additionally, the development of low-pollution and low-consumption green industries, the adoption of new water pollution mitigation technologies, and the use of less-polluting pesticides are all effective water pollution reduction strategies [88].

Enriching plant species, reducing landscape fragmentation, and improving the connectivity of ecological corridors can all help to improve habitat quality [75]. In the central and western regions of the nine cities along the Yellow River, the proportion of industrial and agricultural production activities and construction land is large, and the ecological suitability is low. In the future, disorderly construction land expansion should be avoided, and the emphasis should be shifted from large-scale incremental construction to both stock improvement and quality transformation, as well as incremental restructuring, to prevent further erosion of agricultural and ecological land.

### 4.5. Limitations and Future Work

This study examined the trade-offs and drivers of ecosystem services in typical watersheds in nine cities along the Yellow River in Shandong from a static point in time in 2000, 2010, and 2020, with clear regional spatial and temporal heterogeneity. However, the limitations of this paper are mainly focused on the following aspects: firstly, only four types of services were selected; other ESs in the study area also showed slight changes, which makes it difficult to fully characterize the trade-offs of ecological services in the study area; secondly, subsequent field surveys can be conducted to obtain actual measurement data to modify the assessment model used to obtain more accurate results. Therefore, when studying the relationship between ecosystem services, more field survey data should be used to ensure the accuracy of data, and ultimately to improve ecosystem balance and stability and promote sustainable development.

## 5. Conclusions

This paper analyzed the spatial and temporal evolution of four ecological services (WY, CS, HQ, and FP) in nine cities along the Yellow River in Shandong Province, and used root mean square error analysis to determine the trade-offs between ESs and to measure them. Finally, the ALM model was used with the GWR model to identify the main drivers of trade-offs for ESs across time and space, leading to the following conclusion.

The temporal and spatial changes of the four main ecosystem services in SYRB area were inconsistent, with WY showing fluctuations and large interannual differences, CS and FP showing increasing trends, and HQ displaying decreasing trends. In terms of spatial distribution, the high values of CS, WY and HQ are mainly concentrated in the vicinity of the Taishan Mountains, while the high values of FP are concentrated in the southern and northern regions.

The location characteristics of SYRB’s four ecosystem services and their trade-offs are obvious. FP has absolute regional advantages in the trade-offs in SYRB, while HQ has the smallest regional advantages. First, most of the four ecosystem services show significant trade-offs, but the trade-off between FP and CS is very low. Secondly, the trade-off intensity among the four ecosystem services in SYRB shows significant spatial heterogeneity. In addition, there are differences in the changing trend of the overall trade-off intensity among the four ecosystem services. Overall, the areas with higher total trade-off intensity are concentrated in the west and southwest of the northwest, and FP is more dominant in the trade-offs in areas with higher trade-off strength. Areas with low trade-off intensity are concentrated in the central–eastern region, and most of the areas without prominent ESs are concentrated in the region.

The trade-off intensity of ecosystem services in SYRB is affected by multiple factors, and these factors have changed in the past 20 years. In terms of spatial distribution at the county scale, the area of influence of the dominant driver varies considerably between the different trade-off types. The regions with NDVI, CON and PRE as the leading factors are the largest, and especially the area affected by NDVI is the most. The NDVI in SYRB area has been greatly improved during the study period, especially in the counties in the northeast, where the vegetation coverage has increased and the overall ecological environment is better.

## Figures and Tables

**Figure 1 ijerph-19-15681-f001:**
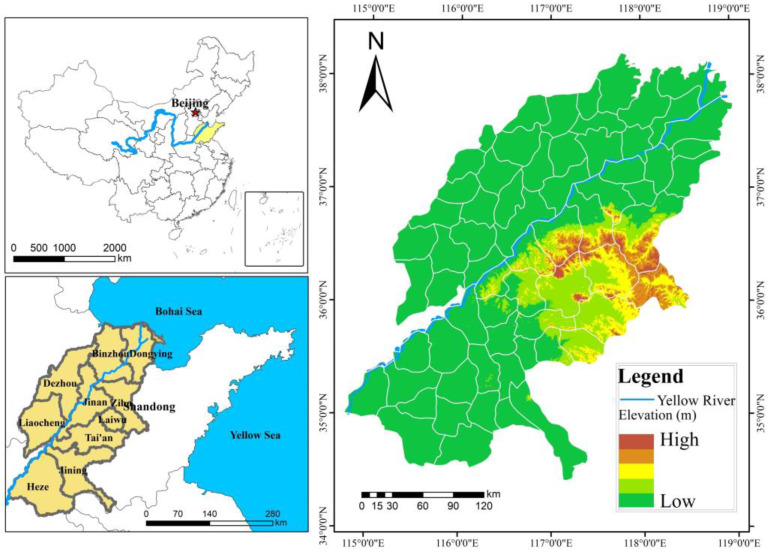
Location and elevation of the study area.

**Figure 2 ijerph-19-15681-f002:**
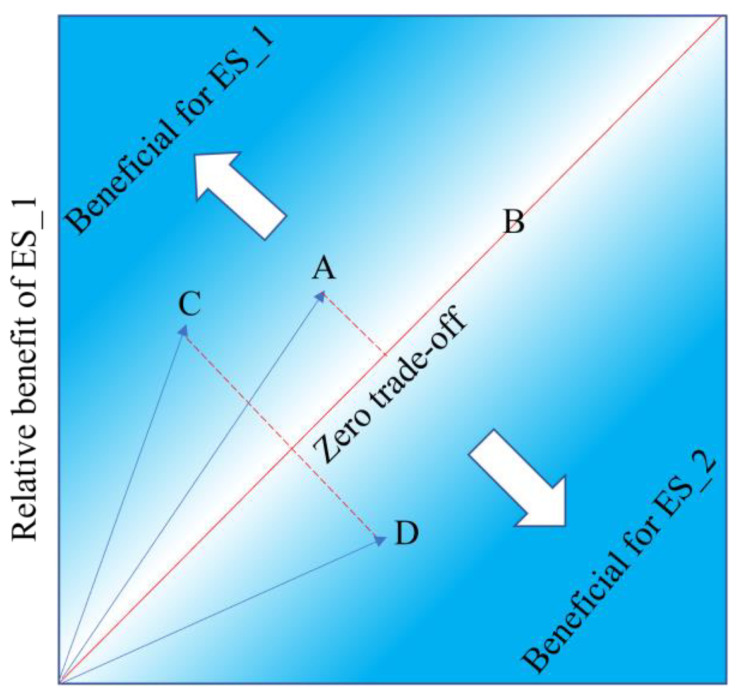
Trade-off between the two ESs is expressed through the root mean square deviation.

**Figure 3 ijerph-19-15681-f003:**
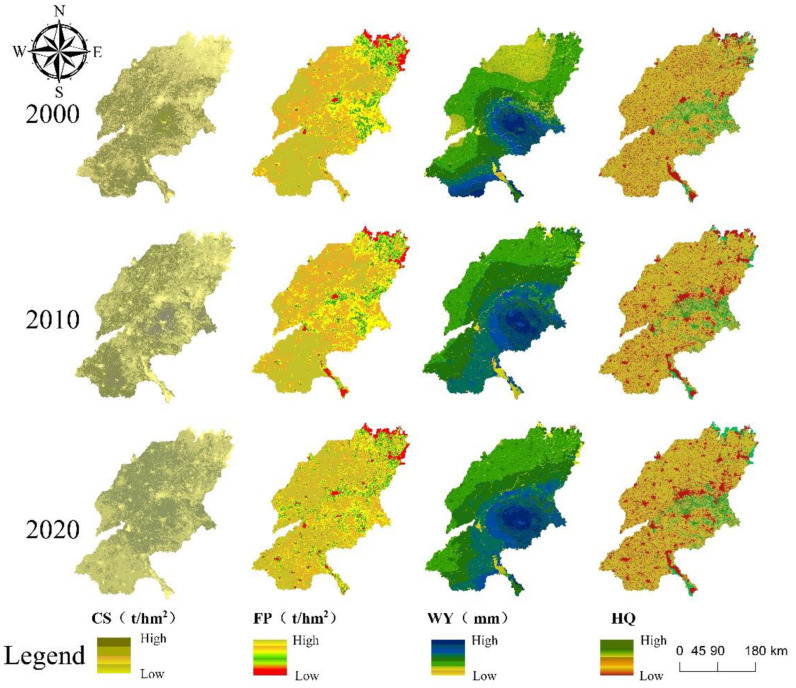
Spatial distribution of four different types of ES during 2000–2020.

**Figure 4 ijerph-19-15681-f004:**
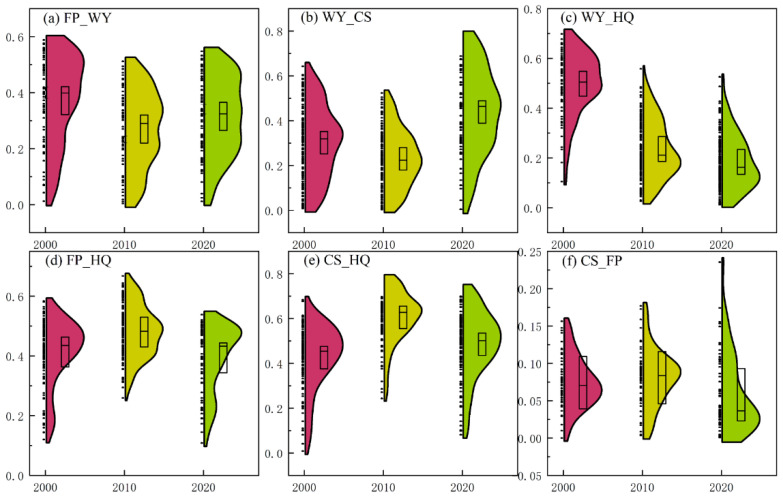
ESs’ trade-offs intensity value domain statistics and error bars (The higher the crest, the more data points contained within the unit range).

**Figure 5 ijerph-19-15681-f005:**
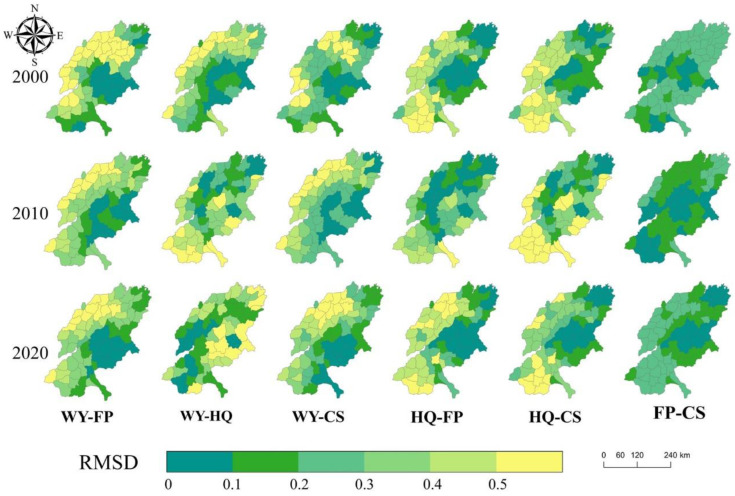
Spatial distribution of the intensity of ecosystem service trade-offs in SYRB.

**Figure 6 ijerph-19-15681-f006:**
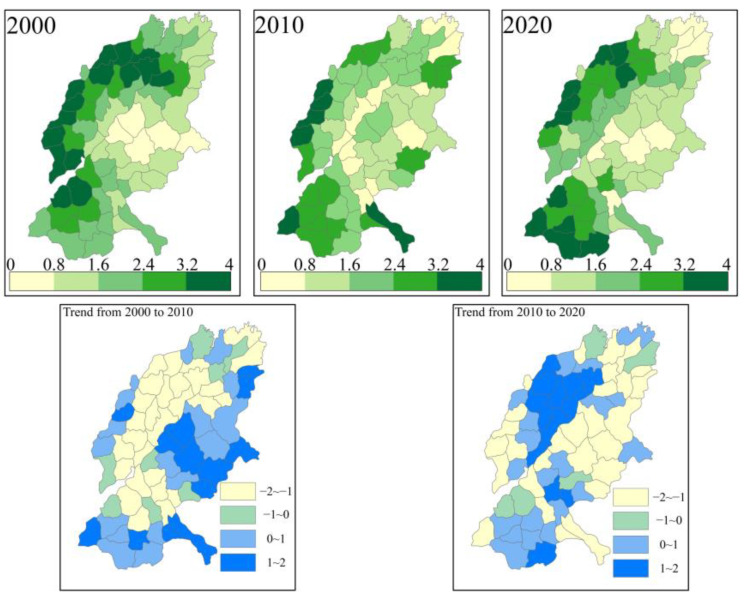
First-row diagram: Spatial distribution of the overall sum of the intensity of different ecosystem service trade-offs in SYRB; Second-row diagram: Trends of the overall sum of the intensity of different ecosystem service trade-offs in different counties in SYRB.

**Figure 7 ijerph-19-15681-f007:**
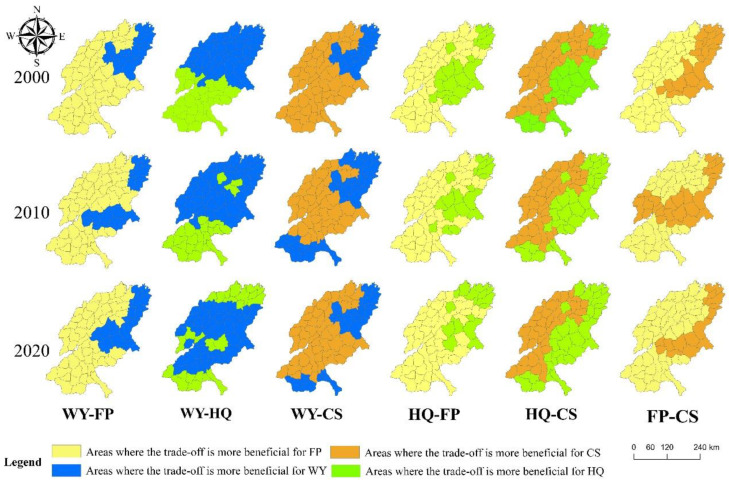
Relative benefits of ESs in the trade-off for different regions (see abbreviations in Figure 2).

**Figure 8 ijerph-19-15681-f008:**
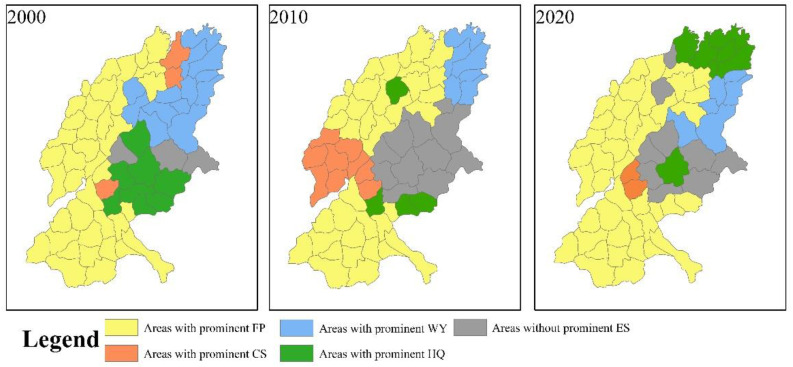
Ecosystem services in which regions are advantaged in trade-offs.

**Figure 9 ijerph-19-15681-f009:**
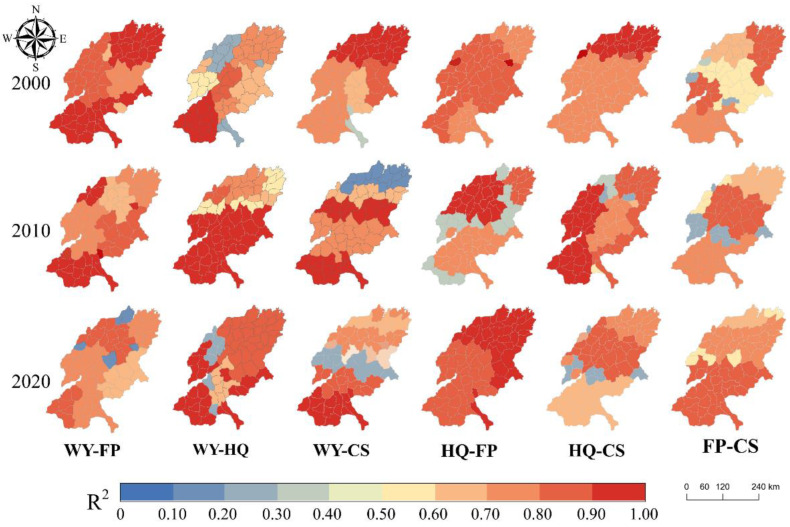
The local R2 of the GWR model.

**Figure 10 ijerph-19-15681-f010:**
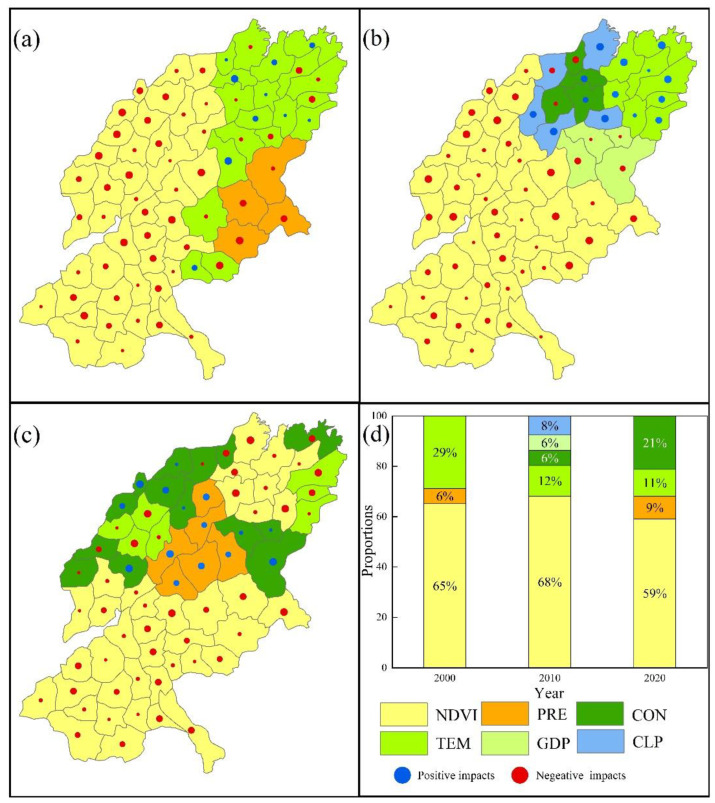
ESs’ trade-off drivers in different cities of FP_HQ ((**a**) in 2000; (**b**) in 2010; (**c**) in 2020). (**d**) Proportion of area accounted for by key ES trade-off drivers in different years (Refer to Table 2 and Table 3 for the meanings of each symbol).

**Figure 11 ijerph-19-15681-f011:**
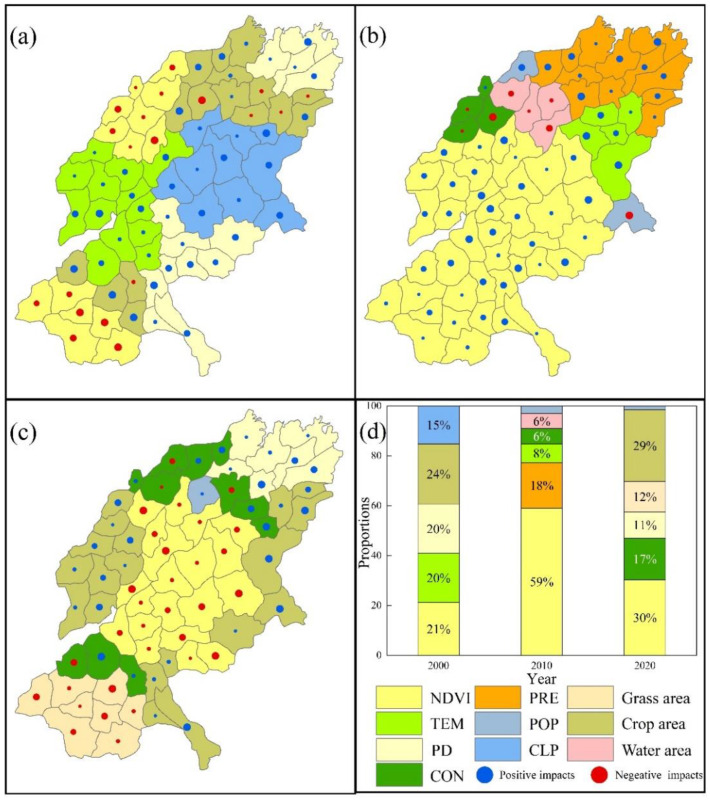
ESs’ trade-off drivers in different cities of FP_CS ((**a**) in 2000; (**b**) in 2010; (**c**) in 2020). (**d**) Proportion of area accounted for by key ES trade-off drivers in different years (Refer to Table 2 and Table 3 for the meanings of each symbol).

**Figure 12 ijerph-19-15681-f012:**
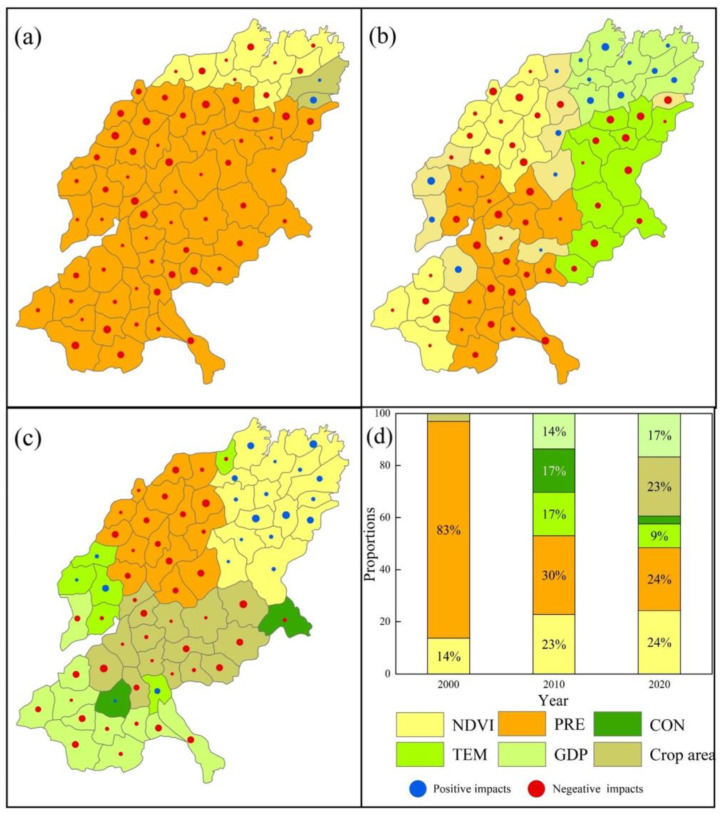
ESs’ trade-off drivers in different cities of FP_WY ((**a**) in 2000; (**b**) in 2010; (**c**) in 2020). (**d**) Proportion of area accounted for by key ES trade-off drivers in different years (Refer to Table 2 and Table 3 for the meanings of each symbol).

**Figure 13 ijerph-19-15681-f013:**
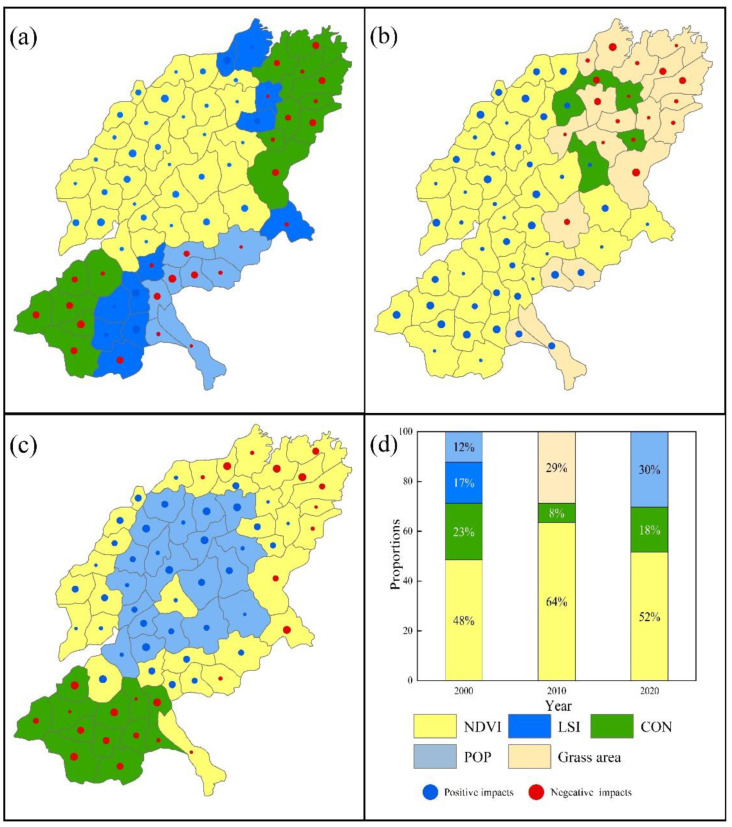
ESs’ trade-off drivers in different cities of CS_HQ ((**a**) in 2000; (**b**) in 2010; (**c**) in 2020). (**d**) Proportion of area accounted for by key ES trade-off drivers in different years (Refer to Table 2 and Table 3 for the meanings of each symbol).

**Figure 14 ijerph-19-15681-f014:**
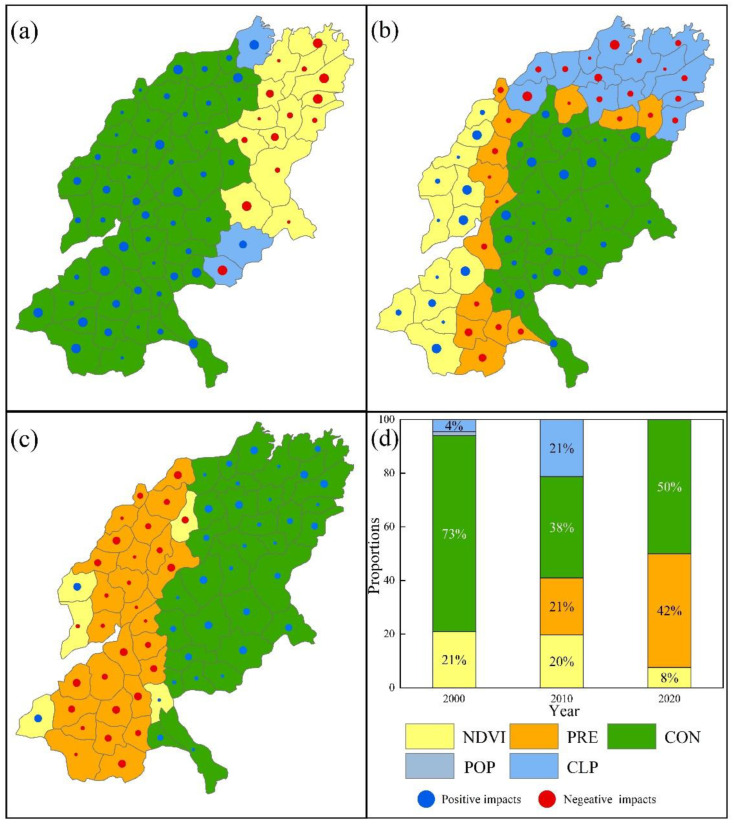
ESs’ trade-off drivers in different cities of WY_HQ ((**a**) in 2000; (**b**) in 2010; (**c**) in 2020). (**d**) Proportion of area accounted for by key ES trade-off drivers in different years (Refer to Table 2 and Table 3 for the meanings of each symbol).

**Figure 15 ijerph-19-15681-f015:**
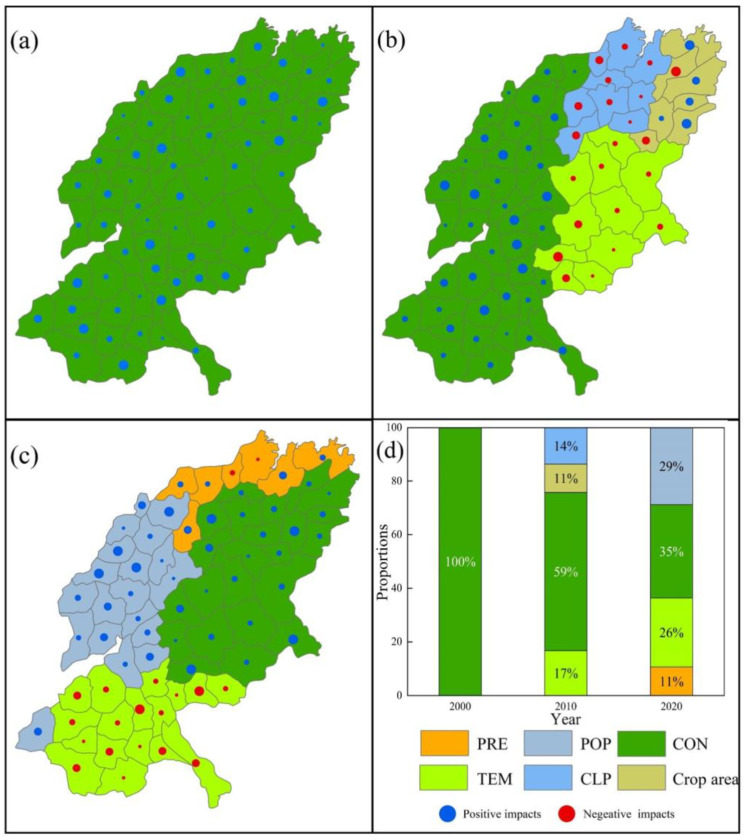
ESs’ trade-off drivers in different cities of WY_CS ((**a**) in 2000; (**b**) in 2010; (**c**) in 2020). (**d**) Proportion of area accounted for by key ES trade-off drivers in different years (Refer to Table 2 and Table 3 for the meanings of each symbol).

**Table 1 ijerph-19-15681-t001:** Methods and main references for evaluating ecosystem services.

Ecosystem Services	Methods	Algorithms	The Meaning of the Symbols in the Algorithms
Habitat Quality	InVEST	$Q_{xj}=H_{j}\left[1-\left(\frac{D_{xj}^{z}}{D_{xj}^{z}+K^{z}}\right)\right],$	$H_{j}$: habitat suitability for habitat type *j*; $D_{xj}$: degree of habitat degradation in pixel x that is in habitat type j; K: half-saturation constant; Z: default parameter of the normalized.
Carbon Storage	CASA InVEST	NPP(x,t)=APAR(x,t)×ε(x,t), $C=C_{soil},$	APAR(x,t): the photosynthetically active radiation absorbed by pixel x in month; ε(x,t): the actual light energy utilization rate of pixel x in month; $C_{soil}:$ Underground carbon stocks of different land use types.
Water Yield	InVEST	$WY_{x}=\left(1-\frac{EET_{x}}{PPT_{x}}\right)PPT_{x},$	$EET_{x}$: annual actual Evapotranspiration; $PPT_{x}$: annual precipitation.
Food Production	Food yield estimation models	$FP_{i}=\frac{NDVI_{i}}{NDVI_{sum}}\times G_{sum},$	$NDVI_{i}$: NDVI value of the i pixel; $G_{sum}$: Total output of food crops; $NDVI_{sum}$: Sum of pixel NDVI values.

**Table 2 ijerph-19-15681-t002:** Description of selected landscape metrics at the landscape level.

Name (Acronym)	Meaning	Equations
Landscape shape index (LSI)	Reflect the shape characteristics of patches in landscape	$LSI=\frac{0.25L}{\sqrt{A}}$, where *L* is the circumference of patches; A is Area of patches.
Contagion index (CON)	Reflects the degree of aggregation of different patch types in the landscape	$CONTAG=1+\sum_{i=1}^{m}\sum_{j=1}^{n}\frac{p_{ij}\ln\left(p_{ij}\right)}{2\ln\left(m\right)}$, where *P_ij_* represents the perimeter of a patch.
Shannon’s diversity index (SHDI)	Reflecting the contribution of rare patches to information	$SHDI=-\sum_{i=1}^{m}p_{i}lnp_{i,}$ where *m* is the number of landscape patch types, and *P_i_* represents the perimeter of a patch.
Patch density (PD)	Reflects the heterogeneity of the landscape per unit area	$PD=\frac{NP}{A},$ where *NP* is the number of patches, *A* is the total area.

**Table 3 ijerph-19-15681-t003:** Description of potential driving factors investigated in this study.

Types	Factors (Unit)	Code
Natural factors	Average precipitation (mm)	PRE
	Average temperature (°C)	TEM
	Normalized Difference Vegetation Index	NDVI
Land-use factors	The total area of cropland (%)	Crop
	The total area of forest land (%)	Forest
	The total area of grassland (%)	Grass
	The total area of water (%)	Water
	The total area of urban land (%)	Urban
	The total area of unused land (%)	Unused
Landscape configuration	Landscape Shape Index	LSI
	Contagion (%)	CON
	Shannon’s Diversity Index	SHDI
	Patch Density (Unit/100 ha)	PD
Socio-economic factors	GDP per unit area (10^4^ RMB/km^2^)	GDP
	Construction land percentage (%)	CLP
	Population density (persons/km)	POP

**Table 4 ijerph-19-15681-t004:** Results of the 2000 automated linear modeling model.

Driving Factors	FP_HQ	CR (%)	FP_CS	CR (%)	FP_WY	CR (%)	CS_HQ	CR (%)	WY_HQ	CR (%)	WY_CS	CR (%)
R^2^	0.73 **		0.76 **		0.57 **		0.83 **		0.61 **		0.62 **	
NDVI	0.62 **	18.3	1.32 **	6.1	0.38 **	3.6	**−0.91 ****	**31.4**	0.32 **	17.1		
PRE	0.11 **	3.1			**−1.11 ****	**53.4**					−1.73 **	26.4
TEM	0.12 *	2.5	0.38 *	2.3	0.17 **	3.1						
SHDI												
CON	−0.43 **	11.3	−1.13 **	5.3			0.52 **	7.1	**0.51 ****	**38.1**	**1.12 ****	**50.7**
LSI					0.61 *	2.4						
PD			−1.04 **	16.8	−1.11 *	1.2	0.82 **	7.3			−3.31 **	7.3
Forest							−0.61 **	16.1				
Grass							−0.31 *	4.1				
Crop	−0.53 **	12.5	**−2.24 ****	**29.7**	0.71 **	21.1						
Urban			−0.92 **	6.7			0.71 **	7.2	0.36 *	14.1		
Water			−1.66 **	18,4								
Unused			−0.72 *	2.5			-					
GDP			−0.41 **	6,3							−0.32 **	3.7
POP					0.27 **	4.1			−0.21 *	13.2	0.501 **	5.1
CLP	**0.63 ****	**37.2**	0.81 *	2.1			0.61 *	5.1	0.21 *	4.21	−0.81 **	8.3

Notes: *, **, mean *p* < 0.05, *p* < 0.01; Bolded font indicates the main drivers of each trade-off, Refer to Table 2 and Table 3 for the meanings of each symbol.

**Table 5 ijerph-19-15681-t005:** Results of the 2010 automated linear modeling model.

Driving Factors	FP_HQ	CR (%)	FP_CS	CR (%)	FP_WY	CR (%)	CS_HQ	CR (%)	WY_HQ	CR (%)	WY_CS	CR (%)
R^2^	0.62 **		0.51 **		0.73 **		0.61 **		0.88 **		0.72 **	
NDVI	**0.69 ****	**37.5**	−0.23 **	26.1	0.46 **	13.2	0.42 **	9.2	0.41 **	6.2		
PRE			−0.17 **	15.9	**−0.71 ****	**35.1**	−0.31 **	11.2	**0.71 ****	**61.3**		
TEM	0.31 **	7.1	−0.21 **	11.1	0.31 **	6.8					**−0.98 ****	**41.6**
SHDI							0.712 *	2.9				
CON	−0.21	5.7	0.19 **	6.1	−0.73 *	9.1	0.11 *	3.1	−0.34 **	4.21	−0.71 *	32.1
LSI												
PD			−0.56 **	6.4			−0.91 **	5.9				
Forest	0.31 **	11.4			−0.61	12.1	**−0.61 ****	**51.1**	−0.71 **	15.2	0.21 *	1.2
Grass	−0.29 **	6.1					−0.31 **	6.1	−0.32 **	6.2		
Crop	0.23 **	4.1	**−0.31 ****	**31.1**					−0.51 *	1.7		
Urban									0.32 **	2.3	−0.61 **	3.1
Water			−0.37 **	3.1								
Unused												
GDP	−0.19 **	4.1			−0.71 **	12.1	−0.81 **	7.1			0.41	2.1
POP			−0.21 *	2.1							0.63 **	0.9
CLP	0.61 **	24.2	0.36 **	4.2					−0.45 **	4.2		

Notes: *, **, mean *p* < 0.05, *p* < 0.01; Bolded font indicates the main drivers of each trade-off, Refer to Table 2 and Table 3 for the meanings of each symbol.

**Table 6 ijerph-19-15681-t006:** Results of the 2020 automated linear modeling model.

Driving Factors	FP_HQ	CR (%)	FP_CS	CR (%)	FP_WY	CR (%)	CS_HQ	CR (%)	WY_HQ	CR (%)	WY_CS	CR (%)
R^2^	0.72 **		0.63 **		0.61 **		0.58 **		0.88 **		0.61 **	
NDVI	**0.71 ****	**31.3**	**0.62 ****	**31.5**	**0.51 ****	**24.1**	0.32 *	4.2	0.31 **	2.9		
PRE	−0.02 **	6.2			1.52 *	11.5	−0.31 **	7.1	0.69 **	31.2	−1.32 **	31.2
TEM	0.23 **	11.3			−0.61 **	16.6			0.02 *	2.4		
SHDI			−0.22 **	5.2			1.21 **	11.2				
CON	−0.12 **	6.3	−0.19 **	12.3	0.72 *	11.3	0.54 **	5.9	**0.32 ****	**41.2**	**0.51 ***	**41.5**
LSI									−0.21 **	1.1		
PD			0.21 *	3.1			−2.91 **	15.3				
Forest	0.21 **	14.1	0.32 **	17.1			**−0.61 ****	**37.1**	−0.52 **	12.4	0.33 **	14.5
Grass	−0.51 **	4.1										
Crop			−0.92 **	23.2	0.51	8.3	−0.27 **	3.1			−0.62 **	3.1
Urban							−0.90 **	7.1	0.51 **	1.9	−0.34 **	1.2
Water											−0.13 **	1.1
Unused									−0.43 *	0.7		
GDP					−0.51 *	9.5					−0.22 **	0.7
POP							0.31 *	2.2	−0.21 *	0.8		
CLP	0.64 *	23.2	−0.43 **	6.2			−0.41 **	3.9			−0.62 **	1.3

Notes: *, **, mean *p* < 0.05, *p* < 0.01; Bolded font indicates the main drivers of each trade-off, Refer to Table 2 and Table 3 for the meanings of each symbol.

## Data Availability

Not applicable.

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
