# Peer review of "Trade-Off Analyses of Multiple Ecosystem Services and Their Drivers in the Shandong Yellow River Basin"

_ijerph, 2022, doi:10.3390/ijerph192315681_

Round 1
Reviewer 1 Report
The article analyzes the evolution of spatial and temporal patterns of certain ecosystem services (Carbon Storage, Food Production, Habitat Quality and Water Yield) in Shandong Yellow River Basin in 2000, 2010 and 2020. To do so, the authors evaluate ESs using the InVEST model, calculate trade-offs between ESs by the root mean square deviation (RMSD), determine landscape spatial heterogeneity with Landscape Pattern Index, combine landscape factors to ESs through Automatic Linear Model and analyze the impact of key drivers for the trade-offs using Geographically Weighted Regression Models.
Although these methods are sufficiently presented and discussed it is given the impression that these are the ultimate tools to obtain reliable results. In my opinion, a reference that the results are subject to the limitations of the methods is needed in ch. 4.4.
A lot of abbreviations are used and can confuse the reader. Consider a table of abbreviations or reexplain an abbreviation in each chapter. Moreover, some symbols are not explained at all (e.g., in table 1). Also, what is “the circumference of plaque” in table 2!
The paper uses reliable data from established databases-sources.
The results are straightforward and properly presented, except for Fig. 4, which emerges directly after a chapter title (L. 374) and nothing is referred in the text for the graphs in this figure. Moreover, Fig. 4 has trade-offs intensity values lower than Fig. 5 (e.g., FP_WY ranges from 0 to 1 in Fig. 5 but from 0 to 0.6 in Fig. 4).
Many figures with results are presented far away from the referring text, making it difficult for the unfamiliar reader to follow. Consider breaking the continuum of Fig. 9 to Fig. 14.
Conclusions are in line to the results in a well-structured manuscript.
Author Response
Response to Reviewer
Dear Reviewer,
Thank you for your letter and the reviewers’ comments concerning our manuscript entitled “Trade-offs analyses of multiple ecosystem services and their drivers in the Shandong Yellow River Basin” (ID2017578). Those comments are valuable and very helpful. We have read through comments carefully and have made corrections.
We would love to thank you for allowing us to resubmit a revised copy of the manuscript and we highly appreciate your time and consideration.
We also appreciate your clear and detailed feedback and hope that the explanation has fully addressed all of your concerns. In the remainder of this letter, we discuss each of your comments individually along with our corresponding responses.
Point 1:
Although these methods are sufficiently presented and discussed it is given the impression that these are the ultimate tools to obtain reliable results. In my opinion, a reference that the results are subject to the limitations of the methods is needed in ch. 4.4.
Response 1:
We are extremely grateful to reviewer for pointing out this problem. We have revised the methods and our ch4.4 again, and we point out the possible errors caused by the lack of field survey data and the possible deficiencies in the selection of ESs, as well as the direction for further research improvement(L836-L851).
Point 2:
A lot of abbreviations are used and can confuse the reader. Consider a table of abbreviations or reexplain an abbreviation in each chapter. Moreover, some symbols are not explained at all (e.g., in table 1). Also, what is “the circumference of plaque” in table 2!
Response 2:
We are grateful for the suggestion. We sorted out the variables and formulas used in the research, and the meanings of related symbols in Table 1 can be found in the same column reference. We have added a column in Table 1 to explain the meaning of the letters in the formula. We have also added a new column in Tables II and III to characterize the meaning of each abbreviation. In the subsequent tables and figures dealing with abbreviations, we have also informed the reader of the location of the meaning of these abbreviations that can be queried in order to make the article clear to the reader. We replace the plaque with patch, which means the perimeter of the patch, and reinterpreted the formula (Table 2).
Point 3:
The paper uses reliable data from established database-sources.
Response 3:
Thank you for pointing out our shortcomings. The data we have taken are from the existing database here, and we have pointed out the limitation section in the article because there are few measured data to validate them(L852-L853). However, in the discussion section, we checked the accuracy of our data against previous studies (CH4.1).
Point 4:
The results are straightforward and properly presented, except for Fig. 4, which emerges directly after a chapter title (L.374) and nothing is referred in the text for the graphs in this figure. Moreover, Fig. 4 has trade-offs intensity values lower than Fig. 5 (e.g., FP_WY ranges from 0 to 1 in Fig. 5 but from 0 to 0.6 in Fig. 4).
Response 4:
Thank you for your careful observation and for spotting our omission. We have added the expression of Figure 4 in section L446-L449, and changed its name to make readers better understand the meaning of the figure. We carefully examined Figure 5 and found that there were some problems with legend in Figure 5. We recheck our data, then redrew Figure 5 and replaced it. We also check other parts of the article related to Figure 5 and replace and change inappropriate results and conclusions (L487-L489; L865-L869). We also checked other remaining figures to ensure their accuracy.
Point 5:
Many figures with results are presented far away from the referring text, making it difficult for the unfamiliar reader to follow. Consider breaking the continuum of Fig. 9 to Fig. 14.
Response 5:
We think that your suggestion of repositioning the figure is also accurate. We have adjusted the position of Figures 9 to 11 respectively to their relevant parts so that they are not far away from the relevant text for the convenience of the reader.
Point 6:
Conclusions are in line to the results in a well-structured manuscript.
Response 6:
We thank you for providing such useful advice! We rechecked the previous results and conclusions, and further refined and summarized these results. Then we revised the previous conclusions to reflect the focus and to make them correspond to the results(L858-L910).
Finally, we read through the full text and corrected some minor mistakes.
Thank you very much for taking the time to help us improve this article, so that it is more substantial in content and more complete in structure. Your careful review has helped to make our study clearer and more comprehensive. It's a big improvement over what we had before. We are very lucky to have your help. Finally, we wish good health to you, your family, and community.
Reviewer 2 Report
1) In introduction(L22-L42), the results need to be more concise to highlight key points.
2) L126, the selection and meaning of the variables can be explained in the methods section, and the introduction section needs to highlight the innovations.
3) L270, the choice of driver variables can be described in more detail, e.g., whether they can be localized to the Yellow River Basin.
Author Response
Response to Reviewer
Dear Reviewer,
Thank you for your letter and the reviewers’ comments concerning our manuscript entitled “Trade-offs analyses of multiple ecosystem services and their drivers in the Shandong Yellow River Basin” (ID2017578). Those comments are valuable and very helpful. We have read through comments carefully and have made corrections.
We would love to thank you for allowing us to resubmit a revised copy of the manuscript and we highly appreciate your time and consideration.
We also appreciate your clear and detailed feedback and hope that the explanation has fully addressed all of your concerns. In the remainder of this letter, we discuss each of your comments individually along with our corresponding responses.
Point 1:
In abstract (L22-L42), the results need to be more concise to highlight key points.
Response 1:
We are extremely grateful to reviewer for pointing out this problem. We have re-capitulated the article, and further refined and summarized our results and conclusions. Then we revised the abstract to reflect and highlight the focus of our study and its significance in the abstract(L13-L48).
Point 2:
L126, the selection and meaning of the variables can be explained in the methods section, and the introduction section needs to highlight the innovations.
Response 2:
We are grateful for the suggestion. We have streamlined this section of the introduction (L108-L117; L141-L174) and introduced it in the section on methods, including the rationale for the selection and the importance of these services in the study area (L232-L277, It should be noted that We are using the revised status of office, so the reference number for this section has not changed). We have also revised the rest of the introduction, again highlighting the innovative points of this article and the significance of the study(L184-L187).
Point 3:
L270, the choice of driver variables can be described in more detail, e.g., whether they can be localized to the Yellow River Basin.
Response 3:
Thank you for pointing out one of our problems, which will be important for our subsequent revisions. First, we combed the relevant literature and carefully considered the actual situation of the study area; Next, we explain in the text the basis for our selection of drivers(L331-L333), and then we add the reference for selecting each driver in Table 3(L372).
Finally, we read through the full text and corrected some minor mistakes. We also checked English language and style, and modified the inaccurate expression.
Thank you very much for taking the time to help us improve this article, so that it is more substantial in content and more complete in structure. Your careful review has helped to make our study clearer and more comprehensive. It's a big improvement over what we had before. We are very lucky to have your help. Finally, we wish good health to you, your family, and community.
Reviewer 3 Report
The authors examined trade-offs among multiple ecosystem services and their driving factors. The description is specific. The topic of the study falls into the scope of the journal. However, the writing needs to be improved a lot. In addition, the innovation point is not very clear.
Major concerns:
Previous studies have already examined trade-offs between ecosystem services and their spatio-temporal variations in different ecosystem types. I am wondering what is the innovation point for this manuscript. Eventhough in the conclusion part, the authors highlighted the importance of considering biological, physical and chemical processes behind the trade-offs between ecosystem services, some analyses were rather confusing. e.g., the trade-off between food production services and carbon storage. This trade-off assumes that better food production services lead to worse carbon storage? In addition, the selection of driving factors should not only be based on statistical analysis, but have clear biological or physical significance. A simple method is to find related references which proved these factors indeed drove the ecosystem services.
The writing must be checked throughout the manuscript. I only gave some examples.
Line 18: ‘And quantitatively measures’.
Line 21: ‘Further anslyzes’
Line 219: ‘above ground and underground’, ‘above-ground’
Some results used present tense, while others used past tense. These errors are too much.
Author Response
Response to Reviewer
Dear Reviewer,
Thank you for your letter and the reviewers’ comments concerning our manuscript entitled “Trade-offs analyses of multiple ecosystem services and their drivers in the Shandong Yellow River Basin” (ID2017578). Those comments are valuable and very helpful. We have read through comments carefully and have made corrections.
We would love to thank you for allowing us to resubmit a revised copy of the manuscript and we highly appreciate your time and consideration.
We also appreciate your clear and detailed feedback and hope that the explanation has fully addressed all of your concerns. In the remainder of this letter, we discuss each of your comments individually along with our corresponding responses.
Point 1:
Previous studies have already examined trade-offs between ecosystem services and their spatio-temporal variations in different ecosystem types. I am wondering what is the innovation point for this manuscript.
Response 1:
We are extremely grateful to reviewer for pointing out this problem. In our initial paper, we did have problems that the expression of innovation points was not clear and the expression of research meaning was not specific enough. We re - refine the content of the article, clear the innovation of this article (L24-L36; L183-L187). We have also combed and improved other parts of the article in order to let readers better understand the main idea of this article.
Point 2:
Even though in the conclusion part, the authors highlighted the importance of considering biological, physical and chemical processes behind the trade-offs between ecosystem services, some analyses were rather confusing. e.g., the trade-off between food production services and carbon storage. This trade-off assumes that better food production services lead to worse carbon storage?
Response 2:
We are grateful for the suggestion. We have fully revised the discussion and conclusion sections, including the limitations of the paper in the discussion. Then, we point out the possible errors caused by the lack of field survey data and the possible deficiencies in the selection of ESs, as well as the direction for further research improvement(L836-L851); And we also made a summary of the results in the conclusion. We rechecked the previous results and conclusions, and further refined and summarized these results. Then we revised the previous conclusions to reflect the focus and to make them correspond to the results(L858-L910). Trade-offs of ecosystem services are not only in the form of one increase and the other decrease, but also in the form of different rates of growth in the same direction (L64-L65), that is, both increase or decrease at the same time, but at different rates of increase or decrease. The trade-off between CS_FP is expressed in this form.
Point 3:
The selection of driving factors should not only be based on statistical analysis, but have clear biological or physical significance. A simple method is to find related references which proved these factors indeed drove the ecosystem services.
Response 3:
Thank you for pointing out one of our problems, which will be important for our subsequent revisions. First, we combed the relevant literature and carefully considered the actual situation of the study area; Next, we explain in the text the basis for our selection of drivers(L331-L333), and we also add the references for selecting each driver in Table 3(L372).
Point 4:
The writing must be checked throughout the manuscript.
Response 4:
Thank you for your careful observation and suggestions. First of all, we have corrected the three mistakes you pointed out, and then we have reconsidered the tense of the full text and corrected the inappropriate places in it.
Finally, we read through the full text and corrected some minor mistakes.
Thank you very much for taking the time to help us improve this article, so that it is more substantial in content and more complete in structure. Your careful review has helped to make our study clearer and more comprehensive. It's a big improvement over what we had before. We are very lucky to have your help. Finally, we wish good health to you, your family, and community.
Round 2
Reviewer 3 Report
The revised manuscript improved a lot. However, the definition of trade-off in this manuscript is rather confused. The authors defined the trade-offs as different changing rates of ecosystem services. I think this is not suitable. If the authors insist, please give a clear definition of trade-off in the introduction part. The knowledge gap (or innovation point) is still not clear. In the response letter, the authors mentioned this has been revised. But I couldn't capture the revision.
